# Programmed biomolecule delivery to enable and direct cell migration for connective tissue repair

Feini Qu[1,2,3], Julianne L. Holloway[2,3], John L. Esterhai[2], Jason A. Burdick[2,3] & Robert L. Mauck[1,2,3]

Dense connective tissue injuries have limited repair, due to the paucity of cells at the wound site. We hypothesize that decreasing the density of the local extracellular matrix (ECM) in conjunction with releasing chemoattractive signals increases cellularity and tissue formation after injury. Using the knee meniscus as a model system, we query interstitial cell migration in the context of migratory barriers using a novel tissue Boyden chamber and show that a gradient of platelet-derived growth factor-AB (PDGF-AB) expedites migration through native tissue. To implement these signals in situ, we develop nanofibrous scaffolds with distinct fiber fractions that sequentially release active collagenase (to increase ECM porosity) and PDGF-AB (to attract endogenous cells) in a localized and coordinated manner. We show that, when placed into a meniscal defect, the controlled release of collagenase and PDGF-AB increases cellularity at the interface and within the scaffold, as well as integration with the surrounding tissue.

[1] McKay Orthopaedic Research Laboratory, Department of Orthopaedic Surgery, Perelman School of Medicine, University of Pennsylvania, Philadelphia, PA 19104, USA. [2] Translational Musculoskeletal Research Center, Corporal Michael Crescenz VA Medical Center, Philadelphia, PA 19104, USA. [3] Department of Bioengineering, University of Pennsylvania, Philadelphia, PA 19104, USA. Feini Qu and Julianne L. Holloway contributed equally to this work. Correspondence and requests for materials should be addressed to J.A.B. (email: burdick2@seas.upenn.edu) or to R.L.M. (email: lemauck@pennmedicine.upenn.edu)

Dense connective tissues of the musculoskeletal system function in demanding biomechanical environments. To withstand repetitive mechanical loads, tissues such as the knee meniscus[1], tendons and ligaments[2], and the annulus fibrosus of the intervertebral disc[3], develop dense and highly aligned collagen-rich extracellular matrices (ECMs). Although this structure-function relationship is optimal for load transfer in the healthy state, injuries compromise this fiber-reinforced tissue architecture and reduce load-bearing capacity[4, 5]. Moreover, mature dense connective tissues exhibit impaired healing[6–8], predisposing the joint to further degenerative changes. This limited healing is commonly attributed to the poor vascular supply and low cellularity of adult tissues, which restrict the local supply of cells that can participate in repair. Additionally, recent evidence suggests that interstitial migration of cells to the injury site is impeded by the adult microenvironment, which is both denser and stiffer than the fetal microenvironment[9]. Furthermore, when injuries occur in the intra-articular environment, the surrounding synovial fluid suppresses clot formation[10], limiting the development of a provisional matrix and abrogating the establishment of local gradients in growth factors and cytokines that would normally recruit reparative cells to the injury site.

A number of studies have recognized limited cell and vascular penetration of the meniscus as barriers to endogenous repair, especially in the relatively hypocellular and hypovascular adult meniscus. To that end, investigators have developed methods for cell delivery to the wound site[11, 12], as well as the introduction of physical conduits[13, 14] or induction of new vascular pathways via the delivery of angiogenic factors to engage the local blood supply[15]. Alternatively, we recently demonstrated that partial degradation of the wound site could enhance cellular migration by reducing the adult ECM stiffness and density to a state more closely approximating the fetal environment[9]. While this finding suggested that adult cells are capable of repair once they reach the wound margin (given the increased integration observed), it did not address the issue of directional migration. Since directional movement of reparative cells into damaged tissue is integral for efficient neotissue formation during healing[4], we hypothesized that the provision of an exogenous chemoattractive gradient might similarly recruit cells to the wound site to accelerate repair, and that this effect would be enhanced with partial matrix degradation.

To translate this concept to practice, we and others have developed a novel multi-fiber electrospun nanofibrous scaffold[16] as a drug delivery vehicle. Aligned fibrous scaffolds mimic the natural organization of native tissue[17], and can be fabricated with multiple discrete fiber components[18]. These distinct fiber fractions can be functionalized with bioactive factors, either adsorbed or annealed to the fiber surface[19–21], blended directly into the fiber[21–25], contained within the core of a hollow co-axial fiber[24, 26, 27], or included within drug-loaded microspheres that are co-electrospun[28]. Depending on the fiber fabrication method and degradative rate, release profiles range from an immediate burst release to a sustained delivery over the course of several weeks.

In this study, we seek to enhance endogenous meniscal repair via the application of a tri-component nanofibrous scaffold releasing multiple factors in a temporal sequence to enhance healing. To enable cell migration, a sacrificial water-soluble poly (ethylene oxide) (PEO) fiber fraction first delivers a burst of matrix-degrading enzyme (collagenase) to the wound interface[9], after which slower-degrading fibers composed of hyaluronic acid (HA)[25, 29], a natural polysaccharide, release a chemoattractant (platelet-derived growth factor-AB, PDGF-AB) to recruit endogenous cells to the injury site. The remaining population of stable poly(ε-caprolactone) (PCL) fibers act as a physical template to provide mechanical integrity and instruction for organized ECM synthesis upon cell arrival[30]. Application of these scaffolds recapitulates the hypercellular healing response exhibited by fetal menisci, an initial step towards repair and regeneration of adult dense connective tissues.

## Results

**PDGF-AB stimulates interstitial cell migration**. To establish PDGF-AB as an effective chemoattractant for meniscal cells in the context of biophysical barriers, the migration of adult meniscal cells was assessed as a function of pore size and PDGF-AB dose using a Transwell migration assay (Fig. 1a). Confocal

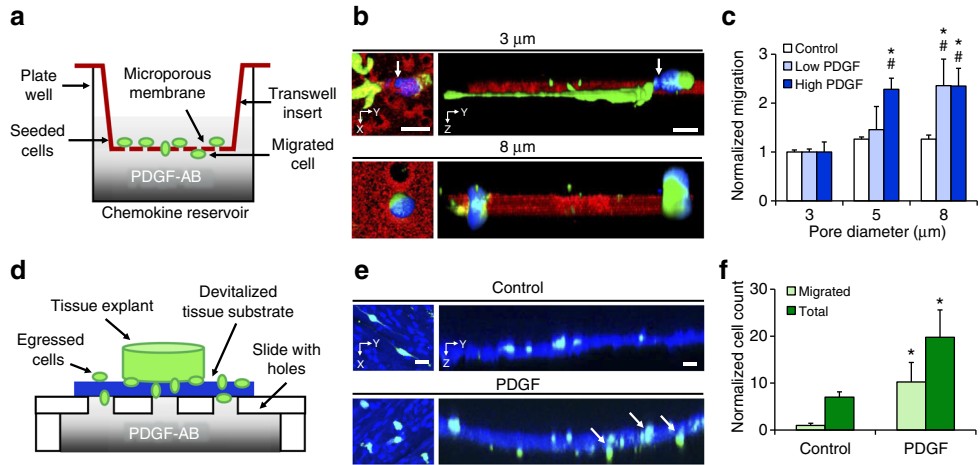

**Fig. 1** PDGF-AB enhances cell migration through microporous membranes and tissue. **a** Schematic of Boyden chamber chemotaxis assay, with platelet-derived growth factor-AB (PDGF-AB) as the chemoattractant. **b** Confocal top–down and cross-sectional views showing adult meniscal cells (green) passing through microporous membranes (red) with 3 and 8 μm diameter pores. Arrow highlights constricted nucleus (blue). Scale = 10 μm. **c** Migrated cell signal intensity normalized to the 3 μm pore group for each media condition ($n = 4$ samples/group, mean ± s.d.). **d** Schematic of a tissue Boyden chamber. A devitalized fetal tissue substrate (blue) is interposed between a glass slide with communicating holes and a chemokine reservoir. Fluorescently labeled cells migrate from an adult tissue explant and through the tissue along the chemotactic gradient, established by the underlying reservoir. **e** Confocal top–down and cross-sectional views showing cells migrating through the tissue barrier. Arrows point to representative migrated cells. Scale = 20 μm. **f** Total number and migrated cells for each media condition ($n = 3$ samples/group, mean ± s.d.). * = $p < 0.05$ vs. Control, # = $p < 0.05$ vs. 3 μm pore

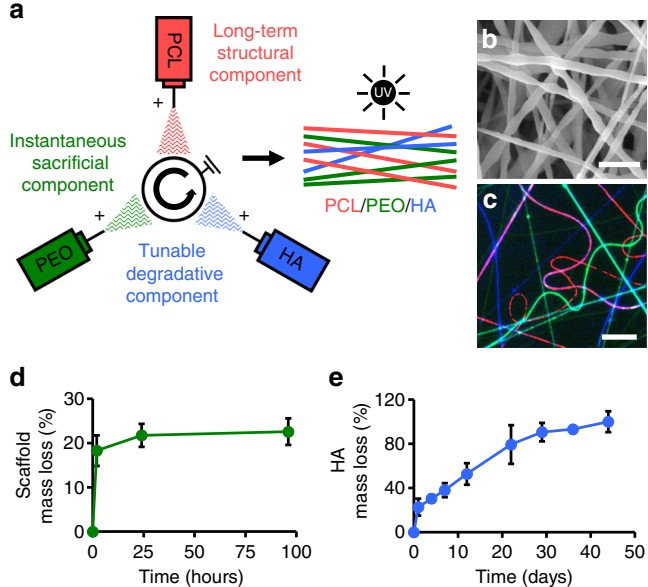

**Fig. 2** Controlled fiber degradation from tri-component nanofibrous scaffolds. **a** Electrospinning schematic with three independent fiber jets collecting simultaneously on a common rotating mandrel. Discrete fiber populations are composed of poly(ε-caprolactone) (PCL), poly(ethylene oxide) (PEO), and hyaluronic acid (HA). **b** SEM micrograph of PCL/PEO/HA scaffold. Scale = 1 μm. **c** Fluorescent image of randomly aligned PCL (red), PEO (green), and HA (blue) fibers. Scale = 20 μm. **d** Bulk scaffold mass loss (% of total) over 96 h (n = 4 samples, mean ± s.d.). **e** HA mass loss (% of total, measured through uronic acid release) over 6 weeks (n = 5 samples, mean ± s.d.)

imaging revealed that meniscus cell nuclei deformed and passed through 5 and 8 μm pores, but could not pass through 3 μm pores (Fig. 1b). In the absence of a PDGF-AB gradient (Control) there was no difference between pore size groups (p > 0.05) (Fig. 1c). With the addition of PDGF-AB to the bottom chamber, cell migration significantly increased as a function of pore size (p < 0.05). The addition of 100 ng/mL PDGF-AB (High PDGF) to the bottom chamber significantly increased migration through both the 5 and 8 μm pores compared to the Control group (p < 0.05). A similar trend was observed for the 50 ng/mL PDGF-AB (Low PDGF) group, though in this case, migration increased for only the 8 μm pores (p < 0.05). In all cases, migration through 3 μm pores was limited, suggesting a minimum pore size for meniscal cell migration that was unaffected by PDGF-AB.

While the above results suggest that a chemoattractant can overcome some steric limitations to migration, these findings were achieved in a non-native artificial environment (i.e., defined circular pores). To determine whether meniscal cells likewise show enhanced interstitial migration towards a PDGF-AB gradient in a tissue-like environment, we developed a novel tissue Boyden chamber (Fig. 1d). When adult tissue explants were placed atop the devitalized fetal meniscal tissue, cells egressed from the tissue explant and adhered, spread, and began migrating into the tissue substrate within 48 h (Fig. 1e). Fetal tissue was used as a substrate for these assays given its lower density compared to adult tissue and the ability for some cell migration to occur through this tissue without collagenase treatment.

In the absence of PDGF-AB in the bottom chamber (Control), relatively few cells migrated through the tissue. The addition of 200 ng/mL PDGF-AB to the bottom chamber significantly increased the total cell number as well as the number of cells that migrated through the tissue compared to the Control group (p < 0.05; Fig. 1f). Together, these data demonstrate that a PDGF-

AB gradient enhances cell migration through membranes of intermediate pore sizes (>5 μm), as well as through a physiologic microenvironment using the tissue Boyden chamber.

**Fabrication and evaluation of tri-component scaffolds.** Towards an implantable system for tissue repair, composite nanofibrous scaffolds containing three distinct fiber fractions were fabricated by electrospinning three polymer sources onto a common rotating mandrel: poly(ε-caprolactone) (PCL), poly(ethylene oxide) (PEO), and hyaluronic acid (HA; Fig. 2a). SEM revealed smooth and uniform nanofibers, with an average fiber diameter of 308 ± 143 nm (Fig. 2b). Images of fluorescently labeled fibers showed the presence of three discrete populations within the scaffold (Fig. 2c). The target composition of scaffolds was approximately 50% PCL, 35% PEO, and 15% HA by mass based on flow rates during fiber production (Supplementary Fig. 1a).

Degradation of each fiber fraction was time-dependent and occurred independently of one another. After 2 h in PBS, 18% of the total scaffold mass was lost, representing the initial loss of the PEO via dissolution and diffusion of uncrosslinked HA from the network (Fig. 2d). After 24 h, mass loss reached 22% and remained constant over the next few days, indicating that the majority of PEO had been removed from the scaffold. In contrast, degradation of the HA component showed a slower and more linear profile, with ~80% HA mass loss over 22 days (Fig. 2e). HA was undetectable after 44 days, resulting in a scaffold that was comprised primarily of PCL at this later time point. Scaffold tensile modulus decreased from an initial value of 7.2 MPa by ~51% immediately after hydration and remained constant thereafter for up to 56 days in PBS, similar to the modulus obtained after hydration and HA removal with hyaluronidase (HASE) digestion. This suggests that the mechanical properties are largely dictated by the PCL fiber fraction and that the slower degradation of HA does not substantially impact mechanics over time (Supplementary Fig. 2).

**High porosity scaffolds expedite cell infiltration.** To assess how endogenous cells interact with these scaffolds, adult meniscal tissue explants were placed atop the scaffolds, and emerging cells adhered, spread, and infiltrated into nanofibrous substrates within 72 h (Fig. 3a, b). Analysis of cell infiltration into high porosity (PCL/PEO) and low porosity (PCL/PEO/HA) scaffolds revealed that infiltration depth was influenced by nanofiber density. The inclusion of HA fibers reduced the pore area fraction to less than half of the PCL/PEO substrate, which contained instead an additional sacrificial PEO fraction (p < 0.05) (Fig. 3c). These low porosity scaffolds had a lower percentage of intermediate-sized pores (>5 μm diameter) compared to high porosity scaffolds (Supplementary Fig. 3b). Consequently, infiltration depth was reduced for cells on low porosity scaffolds compared to cells on high porosity scaffolds (p < 0.05) (Fig. 3d). When HA fibers were selectively removed by digestion with hyaluronidase (PCL/PEO/HA + HASE), the pore area fraction and intermediate-sized pore percentage were restored, and cells migrated into these scaffolds to a similar extent as in high porosity scaffolds. Nanofiber density also modulated nuclear morphology during migration, where nuclei in high porosity scaffolds were narrower and more irregular than those in low porosity scaffolds (Supplementary Fig. 3). Smaller nuclear width was correlated with increased infiltration depth, suggesting that nuclei were actively deforming as cells squeezed through the nanofibrous matrix. These results indicate that nuclear deformation is required for cell mobility through dense nanofibrous

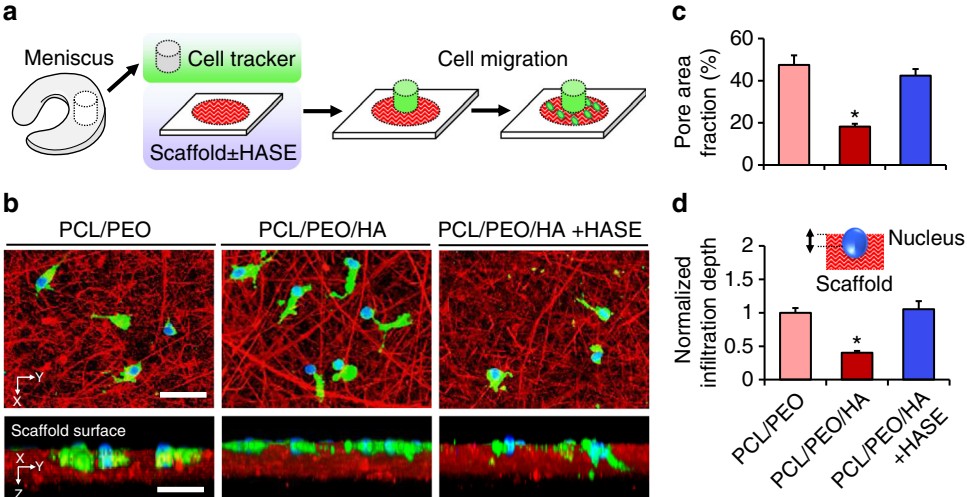

**Fig. 3** High porosity scaffolds enhance cell infiltration. **a** Experimental schematic showing fluorescently labeled tissue explant placed atop nanofibrous substrates with or without hyaluronidase (HASE) treatment. **b** Confocal top–down and cross-sectional views of cells (green) and corresponding nuclei (blue) within nanofibrous substrates (red). Scale = 50 μm. **c** Scaffold pore area fraction ($n = 3$ samples/group, mean ± s.d.). **d** Nuclear infiltration depth with schematic inset, normalized to the highly porous PCL/PEO control ($n = 20$ cells/group, mean ± s.e.m.). * = $p < 0.05$ vs. all other groups

microenvironments, and that there is a minimum porosity threshold that must be reached before cells are able to migrate.

**Controlled biofactor release from tri-component scaffolds.** Having established that cells can migrate into scaffolds, we next focused on recruitment of cells to the wound interface via controlled digestion of the surrounding tissue and the establishment of a chemoattractive gradient towards the implant site. Collagenase and PDGF-AB were successfully incorporated and released from the PEO and HA fiber fractions, respectively, with biofactor release occurring independent of one another and corresponding with the dissolution and degradative rates of the respective fiber fractions (Fig. 4). As expected, there was an instantaneous burst release of collagenase from the water-soluble PEO fiber fraction, where 80% was released within 5 h of incubation in PBS (Fig. 4a). Based on fluorescent activity assays, ~57% of the total released collagenase was active, with $38.4 \pm 3.7$ μg of active enzyme delivered per mg of scaffold (enzyme activity was not detected in control scaffolds) (Fig. 4b). To further demonstrate collagenase activity, cartilage tissue samples were incubated in the presence of scaffolds releasing collagenase (Scaffold + C). Matrix proteoglycan content of these samples was significantly lower compared to tissue incubated with scaffolds without collagenase (Scaffold) and with no scaffold present (Control) (Fig. 4c).

In contrast to collagenase release from PEO fibers, PDGF-AB release from HA fibers showed a more linear profile over the course of 5 weeks (Fig. 4d). Approximately 36 and 67% of PDGF-AB was released after 4 and 12 days, respectively, corresponding to 30 and 52% loss in HA mass due to fiber degradation. PDGF-AB was no longer detectable in the releasate after 44 days. PDGF-AB activity at incremental release time points was assessed using a BrdU proliferation assay (Fig. 4e) and a Transwell migration assay (Fig. 4f), and compared to releasate from scaffolds without PDGF-AB (control, red dashed line). For both adult meniscal cell proliferation and migration, the activity of the 3-day and 7-day releasate from PDGF-AB scaffolds (Scaffold + P) was significantly higher than releasate from scaffolds lacking PDGF-AB ($p < 0.05$).

**Improved tissue repair in an in vivo subcutaneous model.** To evaluate the effect of scaffolds on meniscal repair in vivo, adult

meniscal repair constructs were placed subcutaneously in athymic rats and evaluated histologically at 2 and 4 weeks (Fig. 5a). The implanted scaffolds contained 3 fiber fractions with or without collagenase in the PEO fraction and with or without PDGF-AB in the HA fraction (Fig. 5b). At both 2 and 4 weeks, interfacial cellularity at the center of the repair construct was significantly higher for samples that contained bioactive scaffolds when compared to the empty defect (Empty) and blank scaffold (Scaffold) groups ($p < 0.05$) (Fig. 6 and Supplementary Fig. 4).

At 4 weeks, constructs with scaffolds containing both collagenase and PDGF-AB (Scaffold + C + P) showed the highest cell signal intensity in native tissue within 200 μm of the interface, a significant increase compared to the Empty control, Scaffold alone, and PDGF-AB only scaffold (Scaffold + P) groups (Fig. 6a–c). Collagenase only (Scaffold + C) and PDGF-AB only (Scaffold + P) scaffold groups alone resulted in a two-fold increase in interfacial tissue cellularity when compared to blank Scaffold implants. The PDGF-AB only scaffold had a lower interfacial tissue cellularity when compared to either collagenase-releasing scaffold, suggesting that the ECM presents a formidable barrier to cell-mediated repair. There were no significant differences in tissue cellularity between treatment groups at distances greater than 300 μm, indicating that biofactor release and activity was localized to the defect interface. This demonstrates that the scaffolds can exert a local effect on migration, while not compromising the overall integrity of the tissue.

Despite the complete loss of PEO and partial loss of HA at 2 weeks, few cells were observed within the scaffold center for all groups (data not shown), suggesting that scaffold porosity was too low at this early time point to allow effective cell infiltration. By 4 weeks, however, cells were present within scaffolds from all groups, likely facilitated by the HA fiber degradation over this time period as well as additional time for colonization. While cells were evenly distributed throughout the entire thickness of collagenase-releasing scaffolds (Scaffold + C and Scaffold + C + P), cell ingress was limited to the periphery of scaffolds lacking collagenase (Scaffold and Scaffold + P). When this was quantified, the Scaffold + C + P group had the greatest number of cells within the scaffold center ($p < 0.05$) (Fig. 6d).

Matrix degradation and cell-mediated matrix remodeling at the tissue-scaffold interface were most evident in the collagenase-releasing scaffold groups, where a loose network of thin,

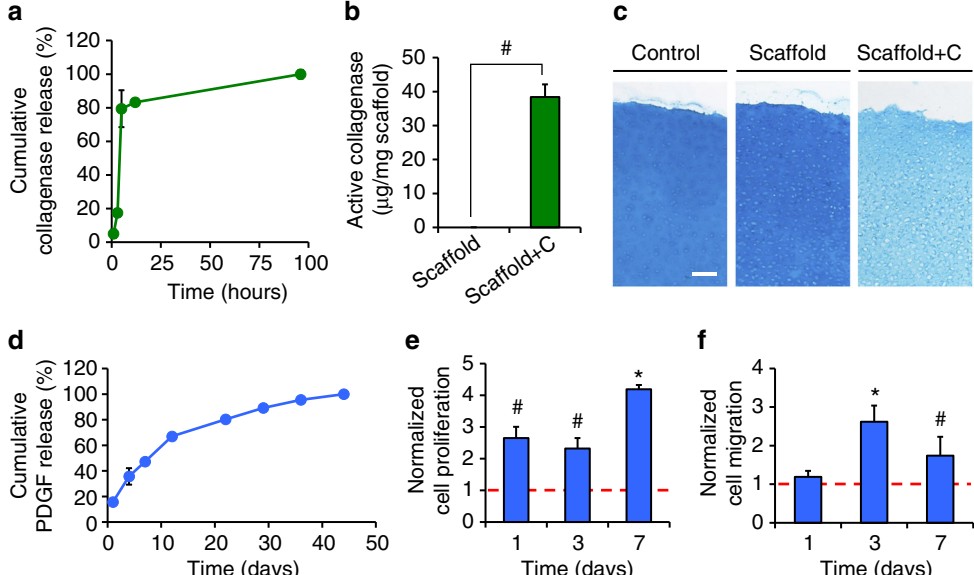

**Fig. 4** Controlled and active biomolecule release from tri-component nanofibrous scaffolds. **a** Cumulative collagenase release (% of total) from scaffolds containing collagenase (Scaffold + C) over 96 h ($n = 5$ samples, mean ± s.d.). Collagenase activity is present in Scaffold + C but not control Scaffold groups ($n = 5$ samples/group, mean ± s.d.), as measured with **b** cleavable peptide substrates or **c** decreased Alcian Blue staining of cartilage explants after 72 h of incubation with scaffolds. Scale = 100 μm. **d** Cumulative platelet-derived growth factor-AB (PDGF-AB) release from scaffolds containing PDGF-AB (Scaffold + P) over 6 weeks ($n = 5$ samples, mean ± s.d.). **e** Proliferation and **f** migration of adult meniscal cells exposed to releasate from Scaffold + P groups at 1, 3, or 7 days of release ($n = 5$ samples/group, mean ± s.d.) normalized to the control Scaffold group (red dashed line). # = $p < 0.05$ vs. Scaffold, * = $p < 0.05$ vs. all other groups

interconnected collagen fibers bridged the scaffold and tissue edges by 4 weeks (Supplementary Fig. 5). Collagen degradation was localized to the interface and did not affect the tissue at distances greater than 300 μm away from the wound site (Supplementary Fig. 6). Additionally, cells and de novo collagen synthesis within the scaffolds were most frequently observed in the groups delivering collagenase (Fig. 7a–d). Cell infiltration distance reached ~85% of the scaffold thickness of Scaffold + C and Scaffold + C + P, compared to ~45% for Scaffold and Scaffold + P ($p < 0.05$) (Fig. 7b). While some collagen deposition occurred at the interface of Empty and Scaffold groups at 4 weeks, the amount was minimal and did not appear well integrated with the surrounding tissue. In contrast, immunostaining revealed that collagen type I and type II were deposited within and around the interface of all three bioactive scaffold groups, evidence that colonization was from endogenous meniscal cells (Fig. 7c). Collagen type II was most abundant within the interior of the dual-factor Scaffold + C + P, with ~2.5 times more staining than the Scaffold group ($p < 0.05$) (Fig. 7d). Interfacial integration for the Scaffold + C + P group (77%) was significantly higher than the control Scaffold (55%) ($p < 0.05$) (Fig. 7e), and marginally greater than Scaffold + C (71%) and Scaffold + P (59%). While fibrous encapsulation was observed at the explant periphery in all groups, the host immune response in this model was minimal (Supplementary Fig. 7).

## Discussion

The ECM of the knee meniscus and other dense connective tissues is a highly functional but physically restrictive 3D micro-environment. Limited porosity and small pore sizes may contribute to the lack of cell-mediated repair in the adult meniscus by obstructing endogenous cell migration to the wound site, even when chemoattractive gradients would favor migration. To address this, we designed scaffolds to direct the coordinated phases of tissue repair by first enabling repair via modulation of

the matrix to permit interstitial cell migration, followed by recruitment of endogenous cells to the wound site via the provision of a chemoattractant (Fig. 8). This was achieved through specific biomaterial design to selectively release biofactors in a temporal sequence during the healing process and to establish a supportive framework for tissue ingrowth.

The healing of adult dense connective tissues is limited in part by the inability of endogenous cells to migrate through the dense and stiff ECM. As with other cell types[31, 32], meniscal cell migration declined with decreasing pore size, a trend that was even more apparent in the presence of a PDGF-AB chemoattractive gradient. Consistent with other reports[33, 34], we found that PDGF-AB was the most effective chemoattractant for meniscal cells through intermediate-sized pores and more importantly, through native tissue. PDGF-AB can also recruit a host of other cell types, including articular chondrocytes, fibroblasts, tenocytes, and mesenchymal stem cells (MSCs)[35–37], rendering our drug delivery system applicable to a wide range of musculoskeletal tissues. In recent studies, we noted that while fetal and adult tissues are both characterized by pores that are on average ~6 to 7 μm in diameter, the pore area fraction in adult tissue (~2%) was five times less than in fetal tissue. The low density of interconnected pores thus appears to restrict cell mobility through adult tissue. A similar finding is noted in scaffolds, where cell migration is improved after enzymatic removal of HA from composite nanofibrous scaffolds, which increased the area fraction of intermediate-sized pores in particular. While both pore size and density affect migration efficiency, size remains the limiting factor, as small transwell pores rendered cells immobile regardless of the biochemical environment.

Since poor interstitial cell mobility and low cell density are independent limitations to wound repair, we sought to increase cellularity at the wound interface using a biphasic approach of matrix modulation and cell recruitment. To localize this effect, electrospun nanofibrous composites were engineered to deliver a

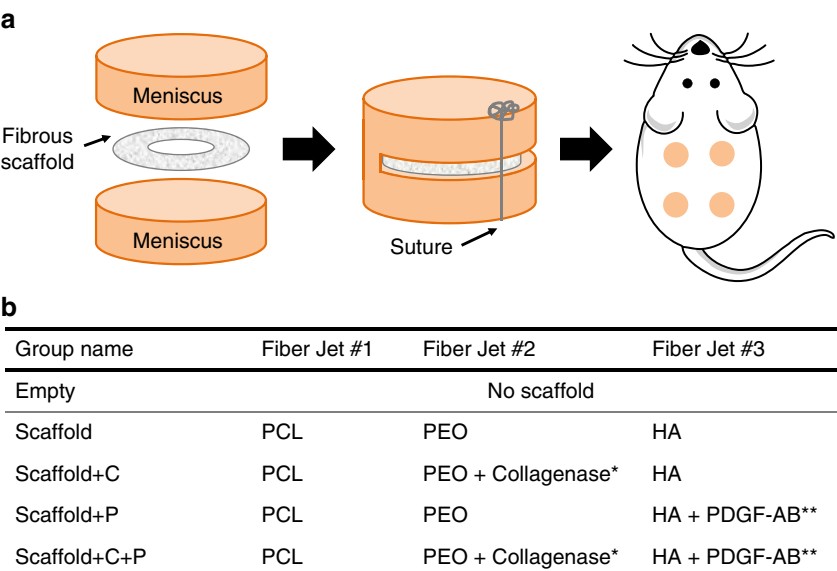

**b**

| Group name | Fiber Jet #1 | Fiber Jet #2 | Fiber Jet #3 |
|---|---|---|---|
| Empty | | No scaffold | |
| Scaffold | PCL | PEO | HA |
| Scaffold+C | PCL | PEO + Collagenase* | HA |
| Scaffold+P | PCL | PEO | HA + PDGF-AB** |
| Scaffold+C+P | PCL | PEO + Collagenase* | HA + PDGF-AB** |

*=173 µg collagenase per scaffold, **=242 ng PDGF-AB per scaffold

**Fig. 5** Implantation groups to assess scaffold-mediated tissue repair. **a** Schematic of a repair construct consisting of a meniscal explant and scaffold that is placed subcutaneously in a nude rat. **b** Fiber and biomolecule composition of each scaffold group. Biomolecule loading (mass per scaffold) is based on electrospinning parameters and scaffold mass

burst of the matrix-degrading enzyme collagenase from water-soluble PEO fibers over the course of 24 h, followed by a sustained release of PDGF-AB from slower-degrading HA fibers over the course of 5 weeks. Targeted removal of collagen and proteoglycans at the interface increased matrix pore size and reduced matrix stiffness, facilitating cell migration and proliferation. When tuned appropriately, the release of collagenase and PDGF-AB increased cell mobility to the wound margin in the short term and resulted in superior scaffold colonization and tissue repair in the long term. While delivery of a chemoattractant (PDGF-BB) from nanofibrous scaffolds has been previously reported[20, 38], this is the first instance, to the best of our knowledge, wherein two factors were delivered in a sequential fashion to enhance both cell mobility and cell migration.

Ideally, the scaffold material would allow for cell attachment, proliferation, and differentiation into a phenotype that culminates in synthesis of the desired ECM components. However, dense nanofiber packing discourages cell infiltration into the scaffold, which is vital for generating functional neotissue. Consistent with other in vitro studies[31, 32, 39], we found that sufficient scaffold porosity and nuclear deformation are crucial for cell migration through a physically restrictive environment. Indeed, the majority of mobile cells are characterized by small and deformed nuclei, suggesting that nuclear deformation is essential for navigating tight interstitia. In this study, modification of HA with hydroxyethyl methacrylate groups promoted fiber degradation in aqueous environments, via a hydrolyzable ester group. This resulted in the steady loss of HA from the composite, reaching ~90% loss over 4 weeks. While the addition of HA fibers may impede migration early on, selective removal of HA with hyaluronidase increased the scaffold porosity and cell infiltration, suggesting that this material will permit cell colonization in the long term. This gradual increase in porosity, along with the time required for cells to reach the scaffold, leads to better colonization of the central region of the scaffold at later time points. Our results indicate that a minimum of ~10% area fraction of pores >5 µm is necessary to encourage cell infiltration.

In this study, we also found that cell migration into scaffolds can be expedited by delivering biofactors. Interestingly, while cellular ingress into scaffolds was apparent by 4 weeks in all groups, infiltration was most advanced in the collagenase-releasing scaffolds. The fragmented collagen at the wound interface may weaken cell anchorage to the ECM, allowing transient detachment to facilitate migration[40]. Additionally, collagenase and its soluble collagen degradation products may themselves act as chemoattractants[41]. Interestingly, while a PDGF-AB gradient enhanced short-term migration in vitro, it did not significantly improve cell infiltration of scaffolds at 4 weeks when delivered alone. Since we did not quantify the spatiotemporal concentration of PDGF-AB, it is unclear whether a chemoattractive gradient existed within the scaffold to direct migration into the scaffold interior. It is also possible that sustained delivery of a high dose of PDGF-AB, a mitogen[33, 42] and stimulator of matrix synthesis[42, 43], resulted in cell proliferation at the interface instead of migration into the scaffold[44]. Alternatively, without modification of tissue pore size, PDGF-AB may have been insufficient to overcome the migratory barriers imposed by the dense ECM at the wound interface. While the origin of these reparative cells requires further elucidation, given that much of the newly deposited tissue is collagen type II positive, we posit that the colonizing cells are either native meniscus cells or endogenous meniscus progenitors that have adopted a fibrochondrogenic phenotype.

Future work will focus on evaluating the long-term anabolic effects of PDGF-AB and tailoring its delivery profile to optimize cell homing after matrix degradation in vivo. Other chemoattractants with little mitotic secondary effects, such as stromal cell-derived factor-1α (SDF-1α)[45], will also be explored. To enhance the quantity and quality of matrix produced by the recruited cells, we are incorporating microspheres[28] into the PCL fiber fraction to slowly release transforming growth factor-β3 (TGF-β3), a potent fibrochondrogenic growth factor[46], and lysyl oxidase, an enzyme that crosslinks and stabilizes nascent collagen fibrils[47]. Lastly, while the subcutaneous implantation model allows short-term assessment of biofactor delivery in an in vivo setting, it does not recapitulate the synovial joint in terms of biochemical milieu, cellular constituents, immune response, and dynamic mechanical loading, all of which could affect wound repair. Ultimately, this

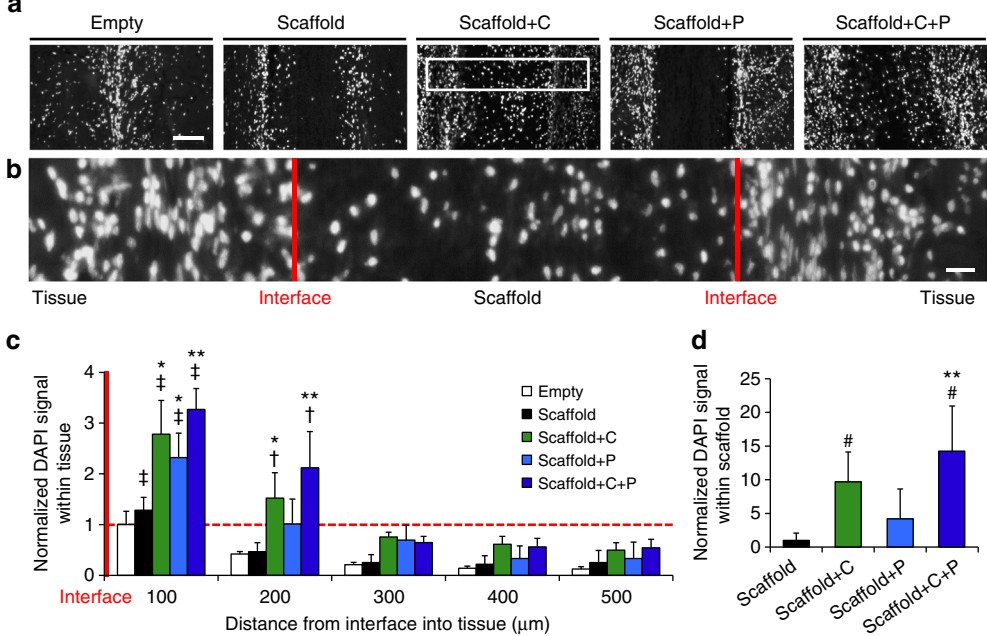

**Fig. 6** Scaffolds releasing collagenase and PDGF-AB increase interfacial cellularity. **a** Images of DAPI-stained nuclei (white) at the center of repair constructs after 4 weeks of subcutaneous implantation. Interface is in the middle. Scale = 100 μm. **b** Magnified image from white box in **a**. Red lines indicate tissue-scaffold interfaces. Scale = 20 μm. **c** Average cell signal intensity with respect to distance from the interface, normalized to the Empty value at the interface and **d** average cell signal intensity within the scaffold center (300 μm thickness), normalized to the control Scaffold group ($n$ = 4–5 samples/group, mean ± s.d.). † = $p < 0.05$ vs. distances ≥ 300 um, ‡ = $p < 0.05$ vs. all other distances, # = $p < 0.05$ vs. Scaffold, * = $p < 0.05$ vs. Empty and Scaffold, ** = $p < 0.05$ vs. Empty, Scaffold, and Scaffold + P

novel biomaterial must be evaluated in an orthotopic large animal model of meniscal injury to validate its long-term therapeutic efficacy.

In summary, this manuscript details the development and application of a versatile biomaterial platform that maximizes cell recruitment while minimizing damage to the tissue. It is important to note that both in vitro and in vivo findings indicate the use of a chemoattractant to increase cell migration is not sufficient without also creating a local ECM microenvironment that is amenable to migration. This conclusion has significant implications for developing future tissue engineering-based therapeutics for dense tissues, where special attention should be focused on creating an environment with the appropriate porosity for cell migration. Furthermore, the proposed system will enable the precise temporal delivery of additional therapeutics, such as anabolic growth factors and anti-inflammatories, which could be used to promote matrix synthesis and inhibit matrix catabolism over the long-term. Through coordinated reprogramming of both cells and matrix, this technology provides a translational framework with which to treat many common musculoskeletal injuries.

## Methods

**Cell isolation and culture.** Fetal (late 2nd–3rd trimester) and adult (20–30 months) bovine stifle joints were sterilely dissected and the knee menisci were removed. To isolate cells, bodies of medial menisci (red/white central zone) were minced, placed onto tissue culture plastic (TCP), and incubated at 37 °C in a basal media (BM) consisting of Dulbecco's modified Eagle's medium (DMEM; Sigma-Aldrich) with 10% fetal bovine serum (FBS) and 1% penicillin/streptomycin/fungizone (PSF). Cells that egressed from the minced tissue onto TCP were expanded to 80% confluency before being detached with 0.25% trypsin-EDTA (Passage 1; P1). Prior to chemotaxis studies, cells were starved overnight in low-serum media (LM; DMEM with 1% FBS and 1% PSF) to minimize exposure to serum.

**In vitro cell migration.** A 96-well Transwell migration assay kit (Chemicon QCM 96-well Migration Assay, EMD Millipore) was used according to manufacturer's instructions to evaluate cell migration as a function of membrane pore size and

chemoattractant (human recombinant platelet-derived growth factor-AB, PDGF-AB; Prospec Bio) concentration (Fig. 1a). 0 (Control), 50, or 100 ng/mL PDGF-AB (Low or High PDGF) in LM was added (150 μL) to the bottom chamber of the wells and 50,000 P1 meniscal cells labeled with 5-chloromethylfluorescein diacetate (CellTracker Green; Thermo Fisher Scientific) were seeded into the top chamber and incubated for 16 h at 37 °C before quantification. Migration was assessed with various pore sizes (3, 5, or 8 μm diameter, $n$ = 4/group) and the fluorescence signal intensity was normalized to the 3 μm pore group for each condition. After migration, the membranes were fixed in 4% paraformaldehyde and stained with 4',6-diamidino-2-phenylindole (DAPI; Thermo Fisher Scientific) to visualize cell nuclei. Cells migrating through the membranes were imaged (Nikon A1 confocal microscope, ×20 magnification, 1 μm intervals) in the FITC (475–490 nm), DAPI (385–400 nm), and TRITC (545–565 nm) channels to visualize cells, corresponding nuclei, and Transwell membranes (auto-fluorescent in the TRITC channel).

To better understand cell migration through tissue, a tissue Boyden chamber migration assay was developed using tissue (~35 μm thick sections via cryostat microtome) excised from the middle zone of a fetal bovine medial meniscus. Tissue sections were placed onto glass slides over four laser-cut holes (1 mm diameter), such that the predominant collagen fiber direction was parallel to the slide surface. The slides were set atop concave glass slides containing 140 μL of either LM or LM with 200 ng/mL PDGF-AB (PDGF) (Fig. 1d) and secured with stainless steel clips. Adult tissue explants labeled with CellTracker Green were placed atop the tissue sections and incubated at 37 °C for 48 h in LM ($n$ = 3/group). Slides were then fixed and stained as above and imaged with confocal microscopy (×10 magnification, 2 μm intervals) in the FITC and DAPI channels to visualize cells, corresponding nuclei, and the tissue barrier (auto-fluorescent in the DAPI channel). To quantify interstitial migration through the tissue barrier, the total number of cells and the number of migrated cells (those entirely embedded within the tissue or that had emerged onto the opposite side) were counted.

**Nanofibrous scaffold fabrication.** Hydroxyethyl methacrylate modified hyaluronic acid (HEMA-HA) was synthesized as described previously[24, 29] via the reaction of the tetrabutylammonium salt of hyaluronic acid (74 kDa; Lifecore Biomedical) and the acid modification of 2-hydroxyethyl methacrylate. HEMA-HA (referred to as HA for simplicity) was purified via extensive dialysis against deionized water at 4 °C, lyophilized, and analyzed with ¹H NMR to determine a functionalization of 30%.

Nanofibrous scaffolds were synthesized through tri-component electrospinning (Supplementary Fig. 1) with the following components: (1) 14.3 wt% poly(ε-caprolactone) (80 kDa PCL; Sigma-Aldrich) in a 1:1 mixture of dimethylformamide and tetrahydrofuran; (2) 8 wt% poly(ethylene oxide) (200 kDa PEO; Polysciences) in 50% ethanol in deionized water; and (3) 4 wt% HA with 2 wt% PEO (900 kDa

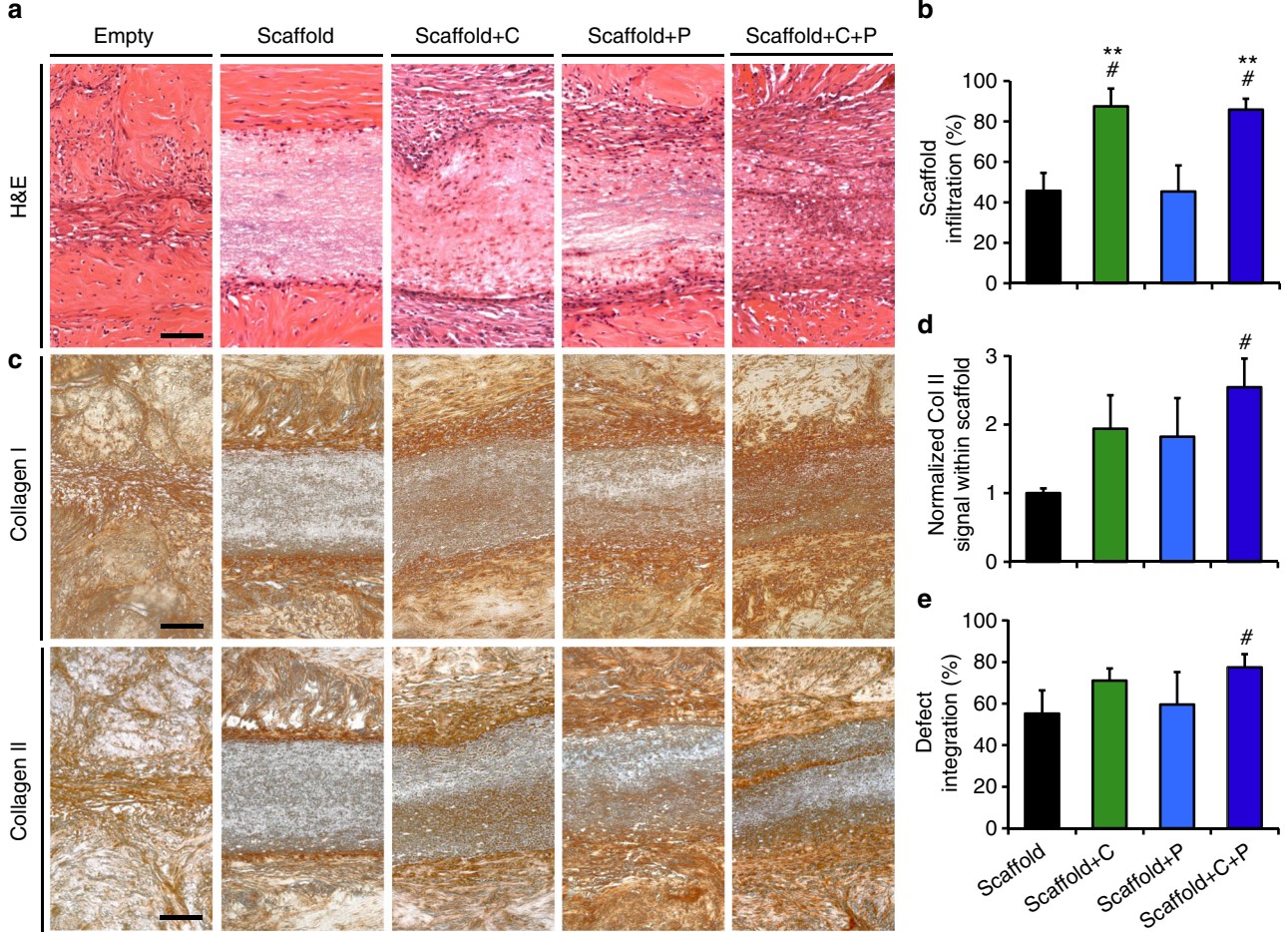

**Fig. 7** Scaffolds releasing collagenase and PDGF-AB enhance tissue integration. **a** Hematoxylin and eosin (H&E) staining of repair constructs show cells, matrix, and nanofibrous scaffolds at the wound site after 4 weeks of subcutaneous implantation. Cells and matrix are observed within the bioactive scaffold groups. Scale = 100 μm. **b** Cell infiltration into scaffold (% of distance into scaffold center) ($n = 4$–5 samples/group, mean ± s.d.). **c** Immunostaining of collagen type I and type II at the wound interface. Scale = 100 μm. **d** Average collagen type II signal within the scaffold, normalized to the control Scaffold group ($n = 3$ samples/group, mean ± s.d.). **e** Defect integration (% integrated tissue at the wound site) for scaffold-containing groups ($n = 4$–5 samples/group, mean ± s.d.). # = $p < 0.05$ vs. Scaffold, ** = $p < 0.05$ vs. Scaffold + P

PEO; Sigma-Aldrich) and 0.05 wt% Irgacure 2959 in deionized water. For scaffolds containing collagenase, 2 wt% collagenase type IV (type IV from *Clostridium histolyticum*, ≥125 collagenase digestion units/mg solid; Sigma-Aldrich) was added to the PEO solution immediately prior to electrospinning. For scaffolds with PDGF-AB, 50 μg/mL PDGF-AB (Shenandoah Biotechnology Inc.) was added to the HA solution immediately prior to electrospinning. Each solution was loaded into individual syringes and placed at equal distances (radially) around a common vertically rotating collection mandrel[18]. Electrospinning was performed using the following parameters for each component: (PCL) +12–15 kV, 2.0 mL/h, 15 cm; (PEO) +12–15 kV, 2.5 mL/h, 15 cm; (HA) +24–28 kV, 1.4 mL/h, 16 cm, corresponding to the applied voltage, polymer flow rate, and spinneret to mandrel distance. Additionally, voltages were applied to the mandrel (−3 kV) and deflector (+4 kV) to increase efficiency of fiber collection on the mandrel. To enable crosslinking of the HA fibers, scaffolds were exposed to 10 mW/cm² of ultraviolet light under nitrogen for 15 min on each side. Scanning electron microscopy (SEM; FEI Quanta 600 FEG Mark II, Penn Singh Center for Nanotechnology Facility) images were used to quantify fiber diameters with NIH ImageJ ($n = 4$ samples, 50 fibers measured per sample). To visualize distinct fiber fractions, the PCL, PEO, and HA solutions were doped with methacryloxyethyl thiocarbamoyl rhodamine B (rhodamine; 0.05% w/v), 5(6)-carboxyfluorescein (0.5% w/v), and DAPI (0.05% w/v), respectively, prior to electrospinning and imaged by fluorescent microscopy.

**Scaffold evaluation.** After electrospinning and crosslinking, tri-component nanofibrous scaffolds (~0.5 mm in thickness) were immersed in phosphate buffered saline (PBS) to assess degradation and biomolecule release. Bulk scaffold mass loss was determined by measuring the dry scaffold weight before and after immersion in PBS for various time points ($n = 4$/group). The amount of uronic acid, a degradation component of HA, in the supernatant was quantified over time using a modified uronic acid assay described elsewhere[48, 49] and compared to

known concentrations of HA ($n = 5$/group). Collagenase release was quantified by measuring the solution absorbance at 280 nm and comparing to a standard curve ($n = 5$/group) and PDGF-AB release was quantified with a PDGF-AB enzyme-linked immunosorbent assay kit (R&D Systems) ($n = 5$/group).

The mechanical properties of tri-component scaffolds at various stages of degradation was evaluated via uniaxial tensile testing. PCL/PEO/HA scaffolds ($30 \times 5 \times 0.5$ mm) were immersed in PBS for 1, 7, 14, 28, and 56 days (Hydrated). Controls included as-spun scaffolds (Untreated) and scaffolds treated with hyaluronidase (HASE; Sigma-Aldrich) for 24 h (6.7 mg/mL in PBS) to remove the PEO and HA fiber fractions. After hydration, scaffolds were lyophilized and the cross-sectional area determined using a custom laser measurement system[17]. Samples were placed in serrated grips, preloaded at 0.5 N, and extended to failure at 0.1% strain per second using an Instron 5848 Microtester and a 50 N load cell[17]. Tensile modulus was calculated from the linear region of the stress (load normalized to cross-sectional area) vs. strain (displacement normalized to gauge length) curve ($n = 4$–6/group).

**Cell infiltration into nanofibrous substrates.** Adult meniscal explants labeled with CellTracker Green were placed on top of PCL/PEO/HA nanofibrous substrates (~20 μm thickness, fluorescently doped with 0.05% wt/v rhodamine) that was previously incubated in PBS with or without 1 mg/mL HASE for 72 h to degrade the HA fiber fraction. Results were compared to a control PCL/PEO scaffold, where the HA fiber component was replaced with an additional PEO fiber component. Explants and scaffolds were cultured for 72 h in BM, followed by fixation and staining with DAPI as described above and confocal imaging (×20 magnification, 1 μm intervals) in the FITC, DAPI, and TRITC channels to visualize cells, corresponding nuclei, and nanofibers.

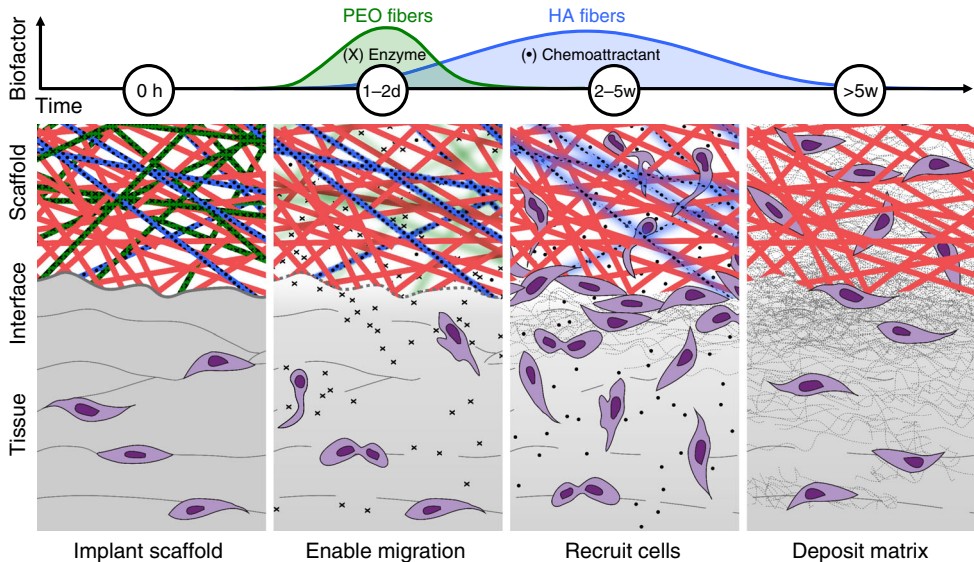

**Fig. 8** Reprogramming the wound interface augments dense connective tissue repair. Water-soluble poly(ethylene oxide) (PEO) nanofibers (green) release a matrix-degrading enzyme (X) in a burst fashion over 24 h, loosening the surrounding matrix and enabling cell migration from the peri-wound matrix to the defect. Controlled release of a chemoattractant (filled black circle) from hyaluronic acid (HA) nanofibers (blue) over 5 weeks recruits additional cells to the interface and into the scaffold. Dissolution of PEO fibers and degradation of HA fibers result in a highly porous scaffold composed of slow-degrading poly(ε-caprolactone) (PCL) nanofibers (red). Enhanced cell colonization and matrix deposition at the interface and within the scaffold result in superior tissue integration and repair

To evaluate cell infiltration into nanofibrous scaffolds, scaffold porosity, nuclear diameter, and nuclear infiltration depth were quantified for each experimental group ($n = 3$ scaffolds/group).

To assess scaffold porosity, the pore size and area fraction were determined using a central $z$- slice from the TRITC channel ($n = 3$ $z$-stacks/group), where areas absent of fibers were considered pores (areas < 20 µm² were considered artifacts and excluded from analysis). The average diameter of discrete pores and the average diameter of nuclei projected in 2D were measured using the image processing software Fiji and the Local Thickness plugin. Cell infiltration into the electrospun fibrous scaffolds was assessed by quantifying nuclear infiltration depth, defined as the distance between the apical surface of the scaffold and $z$-centroid location of the nucleus ($n = 20$ cells/group). The first $z$ location where the fiber signal was detected beneath the cell was defined as the apical surface of substrate. The nucleus $z$-centroid was determined using the BoneJ Particle Analyzer plugin[50]. Cells located within scaffold perforations or defects were excluded from analysis.

**Biomolecule activity**. PCL/PEO/HA nanofibrous scaffolds with or without collagenase in the PEO fraction were incubated in Triton-Tris-Calcium buffer (TTC; 0.05 v/v% Triton X 100, 50 mM tris hydrochloride, 1 mM calcium chloride, pH 7.4) for 2 h with agitation to rapidly dissolve the PEO and release collagenase. The releasate was collected and collagenase activity analyzed by adding 50 µL to a well of a 96-well plate with a matrix metalloprotease-sensitive fluorogenic peptide substrate (Mca-PLGL-Dpa-AR-NH2; R&D Systems) for a final concentration of 10 µM peptide substrate. A fluorescence kinetic study at 320/405 nm excitation/emission with continuous recording over 5 min was performed and compared to collagenase controls to determine activity, and the total amount of collagenase released from each sample was measured using absorbance ($n = 5$/group). As an additional measure of collagenase bioactivity, scaffolds were incubated in the presence of 4 mm cylindrical articular cartilage samples isolated from the trochlear groove of juvenile bovine femurs in BM at 37 °C ($n = 4$/group). After 72 h, the tissue was cryosectioned (16 µm thick) and stained with Alcian Blue to visualize proteoglycan content with optical microscopy.

To assess PDGF-AB bioactivity, meniscal cell (P1) migration and cell proliferation were determined in the presence of releasate from nanofibrous scaffolds. Scaffolds were sterilized under a germicidal lamp, incubated in LM at 37 °C, and releasate collected and frozen after 1, 3, and 7 days for analysis. Meniscal cell migration was quantified using the same 96-well Transwell migration assay kit (Chemicon QCM 96-well Migration Assay, EMD Millipore) described previously ($n = 5$/group). Meniscal cell proliferation was assessed using a bromodeoxyuridine (BrdU) assay kit (Abcam) according to the manufacturer's instructions using a 96-well plate, where 5,000 cells were seeded per well in BM and incubated overnight for adhesion ($n = 5$/group). Afterwards, 100 µL of the releasate was added and cell proliferation was quantified after 24 h.

**In vivo subcutaneous xenotransplant model**. To evaluate the effect of scaffolds on meniscal repair in an in vivo setting, a nude rat xenotransplant model was

employed[9]. All procedures were approved by the Animal Care and Use Committee of the Corporal Michael Crescenz VA Medical Center. Adult bovine meniscal explants (8 mm diameter, 4 mm height, $n = 3$ donors) were incised to create a horizontal defect (Fig. 5a). The defect was either left unfilled (Empty) or filled with one of four nanofibrous scaffolds (Fig. 5b): PCL/PEO/HA (Scaffold), PCL/PEO/HA with collagenase in the PEO fiber fraction (Scaffold + C), PCL/PEO/HA with PDGF-AB in the HA fiber fraction (Scaffold + P), and PCL/PEO/HA with both collagenase and PDGF-AB (Scaffold + C + P). Scaffolds (~0.5 mm thick) were cut to 6 mm diameter with a 2 mm diameter central fenestration to permit tissue-to-tissue contact inside the repair construct and sterilized under a germicidal light prior to use. Biomolecule loading was estimated to be ~173 µg of collagenase and/or ~242 ng of PDGF-AB per scaffold (~2.14 mg). Defects were closed with absorbable sutures before subcutaneous implantation into male athymic nude rats ($n = 5$; Hsd:RH-Foxn1$^{rnu}$, 8–10 weeks old, ~300 g, Harlan). Rats were anesthetized with isoflurane and the dorsal area shaved and scrubbed with ethanol and betadine. Four incisions (1 on the cranial and caudal aspects, on each side of the midline) were made on each animal and a subcutaneous pocket formed using blunt dissection. One randomized repair construct per group was placed in each pocket and the incision was closed with wound clips.

At 2 and 4 weeks, rats were euthanized by CO₂ asphyxiation and the constructs were removed from the subcutaneous tissue. Retrieved samples were paraffin embedded, sectioned to 8 µm thickness, and stained with hematoxylin & eosin (H&E) for tissue morphology, Picrosirius Red (PSR) for collagen, or DAPI for cell nuclei. To assess cellularity, cell signal intensity relative the interface was quantified by converting the DAPI nuclear signal to white (intensity = 255) and the background to black (intensity = 0) using ImageJ, as described previously[9]. Next, the pixel intensity relative to the interface was averaged starting from the interface up to a distance 500 µm perpendicular to the defect. The DAPI signal intensity with respect to location was binned into 100 µm intervals and normalized to the unfilled defect (Empty) value at the interface ($n = 4$–5/group). The average DAPI signal intensity within the scaffold was defined as the average signal of three 100 µm intervals in the centermost portion of the scaffold, spanning a 300 µm thickness. Cell infiltration into the scaffold was measured by dividing the maximum infiltration distance into the scaffold by total scaffold thickness at a randomly chosen point along the scaffold in full H&E cross-sections ($n = 4$–5/group, average of 10 measurements per sample).

Tissue integration was quantified by dividing the cumulative length of integrated segments (based on qualitative adhesion between layers of native tissue and/or scaffold) by the total defect length as measured in full H&E cross-sections. Lastly, immunohistochemistry was performed to identify collagen type I and II deposition[9]. Rehydrated deparaffinized sections were subjected to proteinase K antigen retrieval for 15 min at 37 °C and then incubated with either primary antibodies for collagen type I (10 µg/mL; Millipore) or type II (10 µg/mL; Developmental Studies Hybridoma Bank, University of Iowa) overnight at 4 °C. After washing, signal was detected using an immunoperoxidase secondary detection system (EMD Millipore) per the manufacturer's protocol. ImageJ was used to identify discrete areas that stained positive for collagen type II and the

average signal intensity of 3 centrally located intervals within the scaffold was quantified using the above protocol ($n = 3$/group).

**Statistics**. All statistical analyses were performed using SYSTAT. Significance was assessed by one or two-way ANOVA with Tukey's HSD post hoc testing, where a $p$ value < 0.05 was considered significant. Specifically, for the DAPI analysis, group and bin distance were compared by two-way ANOVA. Data are presented as mean ± standard deviation unless specified otherwise.

**Data availability**. All relevant data are available from the corresponding authors.

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

## Acknowledgements

This work was supported by the National Institutes of Health (R01 AR056624, T32 AR007132, F32 AR063598), the American Orthopaedic Society for Sports Medicine, the Penn Center for Musculoskeletal Disorders (P30 AR069619), and the Department of Veterans' Affairs (I01 RX000174). We gratefully acknowledge Mr Michael Pintauro and Ms. Breanna Seiber for their technical assistance with the tissue Boyden chamber and mechanical testing, respectively.

## Author contributions

F.Q., J.L.H., J.L.E., J.A.B., and R.L.M. conceived and designed the experiments; F.Q. and J.L.H. carried out the experiments and performed the data analysis; F.Q., J.L.H., J.L.E., J.A.B., and R.L.M. co-wrote the paper.

## Additional information

**Competing interests:** The authors declare no competing financial interests.

