## [Peer Review File · Nature Communications]

Reviewers' comments:

Reviewer #1 (Remarks to the Author):

This study investigated the use of nanofibrous scaffold that release molecules to break down ECM and attract cells in a localized manner for improved repair in the meniscus. As noted by the authors, other groups have tried enhancing cell and vascular penetration in dense tissues using similar, and different, techniques. The current work uses a sequential release of collagenase, to loosen the ECM and cell adhesion, followed by PDGF-AB, to attract cells in a directional manner. While this provides some novelty, chemotactic gradients have long been used to attract cells. The "tissue Boyden chamber" provides a bit more potential impact for the field. Reporting the effective pore size used in this test setup (and the in vivo setup) is suggested. The sacrificial fiber strategy is not especially novel, but this is the first time it has been used in this specific way, providing value. Overall, this is a well-written, interesting study. The results are not transformative, but in total, there are plenty of significant aspects presented as part of this work.

Specific comments:

Results, PDGF-AB Cell Migration: The authors state that meniscal cells could not pass through 3 um pores, but then state that there was increased migration through 5 and 8 um pores compared to the 3 um pore insert. Is this really an appropriate comparison? That's similar to saying that there was increased migration compared to a non-porous insert.

What is the effective pore size of the devitalized fetal meniscal tissue?

Results, Fabrication and Evaluation of Scaffolds: With approximately half the scaffold material degrading within a relatively short time, there are concerns about the structural integrity of the scaffold. Is it known how the mechanical properties change at these time points? Is it expected that matrix deposition would happen on the same time scale or is the scaffold over engineered initially to degrade to a mechanical state similar to the tissues around it? This line of questioning is critical to establish the relevance of this study to practical application.

Results, High Porosity Scaffolds: While the findings that cells penetrate more effectively into high porosity scaffolds is to be expected, the authors show nice control in their system for controlling this with different sacrificial fibers and enzymes.

Results, Controlled and Programmed Release of Biofactors: How long is the released collagenase active? A concern is whether this exogenously supplied collagenase will degrade newly synthesized collagen produced by cells infiltrating the scaffold.

Results, Improved Tissue Repair In Vivo: The scaffolds were implanted subcutaneously so experienced no mechanical loading, as it would if repairing a meniscus. Wouldn't the cellular response to this scaffold be completely different in the joint? How useful are the reported findings in understanding what would happen if implanted in sites of matrix-dense connective tissues?

The authors attribute the lack of cellularity at the scaffold center to low porosity but do not explicitly show data supporting this. Was porosity measured through the depth of these scaffolds and did it decrease? If so, why?

Why was collagen type II used for evaluation and not type I? Not a major concern, but just seems a bit odd considering this is not hyaline cartilage tissue.

Methods, In Vitro Migration: Meniscal fiber orientation was directed parallel to the slide surface in the tissue Boyden chamber. Is it expected that cells will be migrating from the top/bottom of the scaffold upon implantation rather than laterally? Are these findings only relevant to certain regions of the native meniscus?

For this and the later Cell Infiltration experiments, what regions of the adult meniscus were explants taken from? Red zone/white zone? Anterior/posterior? Fibrochondrocytes are not a homogenous population in the tissue, so was there any assessment of what cell types actually migrated from the explants?

Methods, Cell Infiltration into Scaffolds: Pore size is measured based on 2D projected area. Will this provide an accurate representation of the 3D pore size? It seems like this approach could dramatically overestimate pore size if the scaffold layers are much thinner than the planar pore size. This question is relevant for all measurements of pore size in this study.

Methods, In Vivo Subcutaneous Model: How are the authors defining "integrated segment?" This is the basis for the tissue integration assessment, but the methods description is quite vague.

Fig. 6c: This graph shows distance from interface into tissue. After a few repeated readings of the body text, it is clear this is reporting accumulation of cells at the interface from the tissue. There may be some confusion from readers because no data are presented regarding cell migration distance into the scaffold (just total numbers). That is often reported for studies using acellular scaffolds, so care should be taken in how this information is presented.

Reviewer #2 (Remarks to the Author):

This manuscript reports the development of a novel tri-component scaffold for meniscus regeneration, capable to release multiple factors. The approach is novel and interesting.

Authors started from the in vitro evidence that PDGF-AB promotes meniscal cell migration and translated this in a tri-phasic scaffold capable to sequentially release different factors and promote repopulation by endogenous cells.

I would recommend to implement the morphologic characterization of the in vivo samples (Figure 7). In particular, it would be great to (i) add a safranin-o staining and col 1 staining in addition to H&E and col2; (ii) add a CD31 (or any other staining for vasculature) staining; (iii) add some cellular staining for MSCs markers in order to understand if the cells migrating to the scaffold have an MSC-like phenotype or are instead mostly differentiated.

Please further comment on possible evolutions of this scaffold and whether adding a trophic signal to the sequence may further help in filling the scaffold with endogenous ECM.

Although suitable for a preliminary evaluation, the nude mice ectopic model does not duplicate the articular environment. Therefore, the degradation/release kinetics can be dramatically different. Please comment on this point.

Although not in the scope of the current paper, an orthotopic animal model seems very important to ultimately assess the potential of this approach. I would suggest outlining such an experiment in the Discussion section, highlighting potential limitations and pitfalls.

Reviewer #3 (Remarks to the Author):

In this study, the a tri-component nanofibrous scaffold was fabricated with 1) a sacrificial PEO outer component that contained collagenase, 2) HA fibers that delivered PDGF, and 3) PCL fibers that gave the scaffold structure to organize incoming cells. The belief is that collagenase will decrease the density of native matrices to release cells that would be attracted into the scaffold via PDGF release. These cells would then organize on the PCL template and form tissue. Extensive work has been done on embedding factors within scaffolds to attract cells and to induce them to produce matrix. Much work has also been done using scaffolds with internal structures that guide cell organization. Combining these with an enzyme that releases cells from the meniscus will be useful for autologous therapies where the scaffold does not contain cells to begin with. To do so would require events to occur with precise timing, and characterization of the kinetics of 1) enzyme release, 2) matrix degradation, 3) PDGF release, 4) cell migration, 5) matrix synthesis, and 6) PCL degradation would allow for a rational description of the effectiveness of this strategy. Times associated with enzyme release, PDGF release, and, to an extent, cell migration and matrix synthesis were examined/manipulated. The extent of matrix degradation was not well-characterized. Other concerns follow below. It is not clear how this strategy would damage the existing meniscus and also other tissues (such as cartilage and ligaments) within the same joint. Collagenase is often used as an osteoarthritis model, and the amount of tissue damage that occur due to collagenase release and the extent of repair that follows need to be characterized since this approach can potentially do harm.

1. Fig 1c - Why normalize migration of 5 and 8um groups to the 3um control group? Shouldn't the PDGF data be normalize within its own group since there may be other issues that give rise to inter-group variability?

2. Fig d – Why devitalized fetal tissue as opposed to adult tissue since presumably the repair would be necessary in adults and not fetuses? I.e., can it be demonstrated in vitro that enzymatically digested adult tissue would provide pores at least 5um in diameter to allow cells to pass through? The strategy of using collagenase to decrease matrix density is predicated on the fact that the meniscus is a dense tissue through which cells have difficulty migrating. If this is the case, adult meniscus tissue, with the relevant density, would be necessary to show that the proposed mechanism is at work.

3. Fig 2d, e and line 144 – After the degradation occurred, were the data measured in wet weight or dry weight? Please provide. For example, after 22 days, were the dry weights of the scaffolds roughly half of the initial dry weights, since all the PEO and 80% of HA would have been gone by then? Normalization of the data in this case is confusing.

4. Fig 3 –Since the HASE treatment occurred for 72 hr in an aqueous solution, did it remove any PEO? How is it certain that the increased cell migration seen in this figure is not attributed to PEO removal as well? Figure 2d shows that most of the PEO (35% of the scaffold) is removed within the first 24 hours, while close to 30% of HA (or 5% of the scaffold) is removed within the first three days (Fig 2e). Since the scaffolds were soaked in HASE for 72 hrs at least 35% of the scaffold would be gone due to PEO degradation. It seems this removal of PEO may have a greater effect on cell migration than HASE digestion (at most 15% scaffold removal since HA is only 15% of the scaffold). Since the authors were able to determine HA mass loss (Fig 2e) as opposed to PEO mass loss, the same method should be used to distinguish whether cell migration of Fig 3 is due to PEO loss or due to HA loss.

5. Similarly, since results for Fig 3 were compared to “a control PCL/PEO scaffold, where the HA fiber component was replaced with an additional PEO fiber component,” (line 417-418), the control scaffold would have approximately 50% of the scaffold removed already after HASE treatment (assuming that control scaffolds were treated the same way as the HASE scaffolds). This would open up much larger

diameter pores than the scaffolds that were implanted into the mice. Should the control have the HA fraction replaced by PCL instead?

6. Fig 4c – how long was this assay (overnight or 24 hr)?

7. Fig 6 – were the meniscus first soaked in tracker dye to distinguish it from mouse cells that could have migrated as well?

8. Fig 6 – Looks like five different statistical tests were performed on this data set. What were the statistical corrections that were made and how? In the Statistics section, it was noted that a Tukey's HSD was performed. Was this done five times on the same data set?

9. Fig 7 – the meniscus contains both collagen I and II. Was collagen I examined?

10. To give a clearer picture of biological variability and sample size, how many mice were used, and how many menisci (from different animals) were used? Samples per group were 4-5?

11. Lines 219-221 – "This demonstrates that the scaffolds can exert a local effect on migration, while not compromising the overall integrity of the tissue." With regard to overall integrity of the tissue, data showing repair or that the surrounding collagen has been damaged/are less dense and how deep this digestion occurs would support the hypothesis and allow a better evaluation of if the treatment is harmful. Supplementary Figure 4 shows something similar to what is requested, but the magnification is too low. Can zoomed-in images be provided of the interface and of the meniscus? What did the interface around the diameter of the scaffold (where it comes in contact with the rat) look like? Were there signs of enzymatic digestion?

12. Supplemental Figure 4 – in addition to zoomed-in images of the meniscus after collagenase digestion, please also provide zoomed-in images of the newly synthesized matrix in the empty and scaffolds.

13. Fig 6d – what is the ROI for "scaffold center"? For example, is it 400-500 um from the interface of the scaffold and the tissue?

14. For the pore size analysis, "areas <20 μm^2 were considered artifacts and excluded from analysis." Were there more of these artifacts for one group/depth versus another? What did the data distribution look like?

Response to Reviewers

Reviewer #1

This study investigated the use of nanofibrous scaffold that release molecules to break down ECM and attract cells in a localized manner for improved repair in the meniscus. As noted by the authors, other groups have tried enhancing cell and vascular penetration in dense tissues using similar, and different, techniques. The current work uses a sequential release of collagenase, to loosen the ECM and cell adhesion, followed by PDGF-AB, to attract cells in a directional manner. While this provides some novelty, chemotactic gradients have long been used to attract cells. The “tissue Boyden chamber” provides a bit more potential impact for the field. Reporting the effective pore size used in this test setup (and the in vivo setup) is suggested. The sacrificial fiber strategy is not especially novel, but this is the first time it has been used in this specific way, providing value. Overall, this is a well-written, interesting study. The results are not transformative, but in total, there are plenty of significant aspects presented as part of this work.

Response: Thank you for your careful reading of our manuscript and positive comments. We have addressed the concerns noted in the response below and in the revised manuscript.

Specific comments:

Results, PDGF-AB Cell Migration: The authors state that meniscal cells could not pass through 3 μm pores, but then state that there was increased migration through 5 and 8 μm pores compared to the 3 μm pore insert. Is this really an appropriate comparison? That's similar to saying that there was increased migration compared to a non-porous insert.

Response: Thank you for pointing out this poor choice of language. We have modified our statement to indicate that the migration through the very small pores is quite minimal, and is not enhanced by the presence of PDGF. Please see revised manuscript page 4, line 118.

What is the effective pore size of the devitalized fetal meniscal tissue?

Response: This is a difficult question to answer. In other work, we measured the apparent pore size of fetal tissue using second harmonic generation imaging (SHG). While the average pore diameter of fetal tissue was similar to adult tissue (approximately 6.6 μm), the pore area fraction in fetal tissue was substantially higher than in adult tissue - approximately 13% (compared to 2% in adult tissue).

Results, Fabrication and Evaluation of Scaffolds: With approximately half the scaffold material degrading within a relatively short time, there are concerns about the structural integrity of the scaffold. Is it known how the mechanical properties change at these time points?

Response: Thank you for your comment. In the past we measured the mechanics as a function of sacrificial components, and showed that so long as the PCL fiber fraction was >40%, then the scaffolds maintained a stable overall structure. In response to this

question, we have since carried out a new study to measure the mechanics of our acellular scaffolds as a function of time, and have added this data as a new supplemental figure (please see Supplementary Fig 2). Our data with this three-fiber fraction scaffold shows that, once hydrated, the mechanics of the scaffold are largely dictated by the PCL fiber fraction. The HA fiber fraction slowly degrades (releasing factors at a controlled rate), but does not substantially impact mechanics after hydration and over time. Discussion of this is now in the results section (please see page 5, line 154).

Is it expected that matrix deposition would happen on the same time scale or is the scaffold over engineered initially to degrade to a mechanical state similar to the tissues around it? This line of questioning is critical to establish the relevance of this study to practical application.

Response: Thank you for your question. In this context, the mechanics of the scaffold are not themselves of major consequence (though they need to have a baseline level of mechanics to allow them to be handled intra-operatively). More importantly, the scaffold is engineered to enable cell colonization and to organize matrix deposition. While the sacrificial PEO fibers are removed within the first 24 hours to facilitate cell migration, significant matrix deposition by infiltrating cells do not occur until 4 weeks post-implantation. In future work, introducing an anabolic growth factor such as TGF-beta may expedite this process.

Results, High Porosity Scaffolds: While the findings that cells penetrate more effectively into high porosity scaffolds is to be expected, the authors show nice control in their system for controlling this with different sacrificial fibers and enzymes.

Response: Thank you for this comment. One of the benefits of this system is the ability to tailor porosity and biofactor delivery in a very precise way.

Results, Controlled and Programmed Release of Biofactors: How long is the released collagenase active? A concern is whether this exogenously supplied collagenase will degrade newly synthesized collagen produced by cells infiltrating the scaffold.

Response: Thank you for this question. Since the PEO dissolved upon hydration, nearly all of the collagenase is released within the first 12 hours. Very little activity is measured subsequent to that time point. Thus, we do not expect that the collagenase should be present any longer, and so should not compromise the nascent collagen production by invading cells, which takes several days to occur. Indeed, the in vivo data seems to bear that out, where new collagen deposition occurred both in and around the scaffold.

Results, Improved Tissue Repair In Vivo: The scaffolds were implanted subcutaneously so experienced no mechanical loading, as it would if repairing a meniscus. Wouldn't the cellular response to this scaffold be completely different in the joint? How useful are the reported findings in understanding what would happen if implanted in sites of matrix-dense connective tissues?

Response: Thank you for this comment. We agree that the dynamic loading environment of a joint is an appropriate next step in evaluating these scaffolds. That said, with respect to mechanical loading, most patients are non-weight bearing for periods of time up to 6 weeks after meniscus repair has been attempted. Therefore, the model in which native

meniscus repair constructs are introduced into the in vivo environment is an appropriate first step, particularly when validating these new formulations. We have updated the text to discuss this model in the context of dense connective tissue repair (please see page 11, line 329).

The authors attribute the lack of cellularity at the scaffold center to low porosity but do not explicitly show data supporting this. Was porosity measured through the depth of these scaffolds and did it decrease? If so, why?

Response: We apologize for the confusion here. Our comment was related to the increasing porosity over time due to the degradation of the HA fiber component. Our data in Figure 3c and 3d indicate that when the HA fiber fraction is present, it reduced pore area fraction and can impede cell infiltration. Once it is removed (via hyaluronidase) infiltration improves along with increases in pore area fraction. In vivo, we suspect that, given the hydrolytic degradation occurring over the course of the first 3 weeks (see figure 2e), porosity of the scaffold gradually increases during the first 2-4 weeks. This, along with the time required for cells to reach the scaffold, leads to better colonization of the central region of the scaffold at later time points. This has been further clarified in the revised text (please see page 10, line 303).

Why was collagen type II used for evaluation and not type I? Not a major concern, but just seems a bit odd considering this is not hyaline cartilage tissue.

Response: Thank you for this question. The meniscus is a fibrocartilage, and so has both type I and type II collagen within, particularly in the inner regions. We stained for both, but focused on type II collagen, as this would indicate that the cells producing the new matrix were of meniscus origin, rather than being cells from the host that had migrated in (which could produce type I collagen). We had originally included picosirius red staining as a supplemental figure (for all collagens) and now specifically include type I immunostaining in Figure 7 (please see Figure 7c).

Methods, In Vitro Migration: Meniscal fiber orientation was directed parallel to the slide surface in the tissue Boyden chamber. Is it expected that cells will be migrating from the top/bottom of the scaffold upon implantation rather than laterally? Are these findings only relevant to certain regions of the native meniscus?

Response: This is a good question. Our technology is meant to address circumferential tears in the meniscus, where the scaffold would be placed with its fiber direction parallel to the prevailing fibers of the native tissue. Thus, migration would occur principally across fibers and into the scaffold, the most difficult configuration for interstitial migration of this kind.

For this and the later Cell Infiltration experiments, what regions of the adult meniscus were explants taken from? Red zone/white zone? Anterior/posterior? Fibrochondrocytes are not a homogenous population in the tissue, so was there any assessment of what cell types actually migrated from the explants?

Response: Thank you for this comment. All donor tissues were isolated from the central region of the meniscus, in the red/white central zone. This information has been added to the methods section. Please see page 11, line 355. With respect to cell type, we did

not further characterize those that entered the wound site, though they are expected to be a heterogeneous population reflecting those available for repair in vivo. Given that these cells produced type II collagen, we expect that the majority of these cells are either meniscus fibrochondrocytes, or endogenous progenitors that have taken on this phenotype.

Methods, Cell Infiltration into Scaffolds: Pore size is measured based on 2D projected area. Will this provide an accurate representation of the 3D pore size? It seems like this approach could dramatically overestimate pore size if the scaffold layers are much thinner than the planar pore size. This question is relevant for all measurements of pore size in this study.

Response: We agree with the reviewer that it is challenging to appropriately denote pore size in a fibrous network. Our approach was to measure the apparent pore area fraction visible under SEM and/or fluorescent imaging as a metric for comparison between scaffold types and with degradation. Our data suggests that this measure correlates with the capacity of a cell to enter the network, and so is a good indicator of permissivity to migration.

Methods, In Vivo Subcutaneous Model: How are the authors defining “integrated segment?” This is the basis for the tissue integration assessment, but the methods description is quite vague.

Response: Thank you for this comment. We have added additional details to the Methods section (please see page 17, line 529). This was based on the apparent adhesion between layers of native tissue and/or scaffold measured on full histological sections stained with H&E.

Fig. 6c: This graph shows distance from interface into tissue. After a few repeated readings of the body text, it is clear this is reporting accumulation of cells at the interface from the tissue. There may be some confusion from readers because no data are presented regarding cell migration distance into the scaffold (just total numbers). That is often reported for studies using acellular scaffolds, so care should be taken in how this information is presented.

Response: Thank you for this comment. We have revised the text in the results section to emphasize the different measures being taken. We also include now in Figure 7 the distribution of cells within the scaffold relative to the scaffold/tissue interface (please see Figure 7b).

Reviewer #2

This manuscript reports the development of a novel tri-component scaffold for meniscus regeneration, capable to release multiple factors. The approach is novel and interesting. Authors started from the in vitro evidence that PDGF-AB promotes meniscal cell migration and translated this in a tri-phasic scaffold capable to sequentially release different factors and promote repopulation by endogenous cells.

Response: Thank you for your careful reading of our work and positive comments.

I would recommend to implement the morphologic characterization of the in vivo samples (Figure 7). In particular, it would be great to (i) add a safranin-o staining and col 1 staining in addition to H&E and col2; (ii) add a CD31 (or any other staining for vasculature) staining; (iii) add some cellular staining for MSCs markers in order to understand if the cells migrating to the scaffold have an MSC-like phenotype or are instead mostly differentiated.

Response: Thank you for these suggestions. Based on this, we have performed type I collagen immunohistochemistry and added the figures to the manuscript, as recommended. We have stained for proteoglycans with Alcian Blue (similar to Safranin O) in previous pilot studies and saw reduced proteoglycans for all groups, including the empty defect control, most likely due to absent mechanical loading in the subcutaneous environment. We have carried out staining for CD31 and other vascular markers and have closely examined our H&E stains, but do not see evidence of vascular structures within the explant tissue, although blood vessels can occasionally be observed in the soft tissue surrounding the explant. As for MSC markers, there are, to our knowledge, no definitive markers that identify the progenitor cell population in the native meniscus. We have previously published that cells within the tissue are multi-potential (adipogenic, osteogenic, and chondrogenic, under defined conditions, please see Mauck et al, Anatomical Record, 2006). Given that much of the newly deposited tissue is collagen II positive, we posit that the colonizing cells are either native meniscus cells or endogenous meniscus progenitors that have adopted a fibro-chondrogenic phenotype. We have added discussion of this to the revised manuscript (please see page 10, line 319).

Please further comment on possible evolutions of this scaffold and whether adding a trophic signal to the sequence may further help in filling the scaffold with endogenous ECM.

Response: Thank you for this comment. Our vision is that the scaffold formulation may be even further refined to address this point. Specifically, we are now working to including TGF-beta and other matrix promoting factors as a very slow release component, which would direct the recruited cells to produce matrix in an expeditious fashion. We have added discussion on this to the revised manuscript (please see page 11, line 326).

Although suitable for a preliminary evaluation, the nude mice ectopic model does not duplicate the articular environment. Therefore, the degradation/release kinetics can be dramatically different. Please comment on this point. Although not in the scope of the current paper, an orthotopic animal model seems very important to ultimately assess the potential of this approach. I would suggest outlining such an experiment in the Discussion section, highlighting potential limitations and pitfalls.

Response: Thank you for this comment. We agree that the ectopic mouse model is a first step in in vivo evaluation, and that an orthotopic placement will be required to fully validate this technology. We have added discussion of this to the revised manuscript per the reviewer's suggestion. Please see page 11, line 332.

Reviewer #3

In this study, a tri-component nanofibrous scaffold was fabricated with 1) a sacrificial PEO outer component that contained collagenase, 2) HA fibers that delivered PDGF, and 3) PCL fibers that gave the scaffold structure to organize incoming cells. The belief is that collagenase will decrease the density of native matrices to release cells that would be attracted into the scaffold via PDGF release. These cells would then decrease the density of native matrices to release cells that would be attracted into the scaffold via PDGF release. These cells would then organize on the PCL template and form tissue.

Extensive work has been done on embedding factors within scaffolds to attract cells and to induce them to produce matrix. Much work has also been done using scaffolds with internal structures that guide cell organization. Combining these with an enzyme that releases cells from the meniscus will be useful for autologous therapies where the scaffold does not contain cells to begin with. To do so would require events to occur with precise timing, and characterization of the kinetics of 1) enzyme release, 2) matrix degradation, 3) PDGF release, 4) cell migration, 5) matrix synthesis, and 6) PCL degradation would allow for a rational description of the effectiveness of this strategy. Times associated with enzyme release, PDGF release, and, to an extent, cell migration and matrix synthesis were examined/manipulated. The extent of matrix degradation was not well-characterized. Other concerns follow below. It is not clear how this strategy would damage the existing meniscus and also other tissues (such as cartilage and ligaments) within the same joint. Collagenase is often used as an osteoarthritis model, and the amount of tissue damage that occur due to collagenase release and the extent of repair that follows need to be characterized since this approach can potentially do harm.

Response: Thank you for your careful reading of our manuscript and comments towards its improvement. With respect to matrix degradation, as noted for motivation of this study (see Qu et al, 2013, Acta Biomaterialia and Qu et al. 2015, Biomaterials), we previously carried out and reported on the effects of collagenase on meniscus tissue matrix in vitro. These data were used to inform the target release levels and duration of release in the engineered multi-fiber technology developed in this manuscript. We also carried out pilot studies using early versions of this scaffold to assess the potential impact of very local collagenase delivery on other structures in the joint. That work showed that the dose we deliver does not have a negative effect on the joint as a whole.

1. Fig 1c - Why normalize migration of 5 and 8um groups to the 3um control group? Shouldn't the PDGF data be normalize within its own group since there may be other issues that give rise to inter-group variability?

Response: Thank you for this comment. We normalized everything to the 3 micron control group to demonstrate that, when pores are small enough, they can limit migration even in the context of a strong chemotactic gradient. If the data were normalized within each group, this important feature would not be observable.

2. Fig d – Why devitalized fetal tissue as opposed to adult tissue since presumably the repair would be necessary in adults and not fetuses? I.e., can it be demonstrated in vitro that enzymatically digested adult tissue would provide pores at least 5um in diameter to allow cells to pass through? The strategy of using collagenase to decrease matrix

density is predicated on the fact that the meniscus is a dense tissue through which cells have difficulty migrating. If this is the case, adult meniscus tissue, with the relevant density, would be necessary to show that the proposed mechanism is at work.

Response: Thank you for this comment. For the 'tissue boyden' assays, we used fetal tissue as a template. This was due to the fact that we've previously shown (see Qu et al.) that the treatment with collagenase converts an adult tissue to one that has matrix mechanics and porosity more akin to fetal tissue. Here, we wanted to assess specifically the effect of PDGF, independent of the collagenase component. For all of the in vivo work presented later, adult tissue was used. The text has been updated to clarify this point. Please see page 5, line 127.

3. Fig 2d, e and line 144 – After the degradation occurred, were the data measured in wet weight or dry weight? Please provide. For example, after 22 days, were the dry weights of the scaffolds roughly half of the initial dry weights, since all the PEO and 80% of HA would have been gone by then? Normalization of the data in this case is confusing.

Response: Thank you for pointing this out. We did measure dry weights to determine mass loss from the scaffolds. We presented the mass in Figure 2d to capture the immediate loss of PEO from the scaffold. We also catalogued mass loss and HA release to the media (using the uronic acid assay) to capture changes in the HA fiber fraction. The uronic acid assay results are shown in Figure 2e, and reported as a percentage of the total HA fiber fraction. The weights and these other independent measure line up with one another.

4. Fig 3 – Since the HASE treatment occurred for 72 hr in an aqueous solution, did it remove any PEO? How is it certain that the increased cell migration seen in this figure is not attributed to PEO removal as well? Figure 2d shows that most of the PEO (35% of the scaffold) is removed within the first 24 hours, while close to 30% of HA (or 5% of the scaffold) is removed within the first three days (Fig 2e). Since the scaffolds were soaked in HASE for 72 hrs at least 35% of the scaffold would be gone due to PEO degradation. It seems this removal of PEO may have a greater effect on cell migration than HASE digestion (at most 15% scaffold removal since HA is only 15% of the scaffold). Since the authors were able to determine HA mass loss (Fig 2e) as opposed to PEO mass loss, the same method should be used to distinguish whether cell migration of Fig 3 is due to PEO loss or due to HA loss.

Response: Since cell infiltration is a function of overall pore fraction, removal of PEO should have a greater impact on cell infiltration compared to removal of HA since there is a larger fraction of PEO relative to HA in our scaffolds. Based on previous work with PCL/PEO scaffolds (Baker et al, 2008, Biomaterials) we know that 30-40% sacrificial fiber fraction is necessary for cell infiltration. Thus, we posit that in scaffold with 85% PCL and 15% HA, HA removal alone would be insufficient to promote cell infiltration. The reviewer is correct that both PEO and HA loss should have occurred with HASE treatment, as opposed to just PEO loss with PBS treatment. However, the effect of HA removal is difficult to isolate since HA digestion also requires hydration of the scaffold. Furthermore, the method used to determine HA mass loss (uronic acid assay) tested for a specific HA degradation component and was not an experiment where HA degradation was isolated from PEO removal.

5. Similarly, since results for Fig 3 were compared to “a control PCL/PEO scaffold, where the HA fiber component was replaced with an additional PEO fiber component,” (line 417-418), the control scaffold would have approximately 50% of the scaffold removed already after HASE treatment (assuming that control scaffolds were treated the same way as the HASE scaffolds). This would open up much larger diameter pores than the scaffolds that were implanted into the mice. Should the control have the HA fraction replaced by PCL instead?

Response: Thank you for this comment. To clarify, loss of either the PEO or the HA fiber fraction will increase porosity. The PEO fiber fraction disappears as soon as hydration occurs, while the HA fiber fraction degrades slowly as hydrolysis of the backbone polymer occurs. One can also add exogenous hyaluronidase to remove these HA fibers, as is shown in figure 3. We did this experiment specifically to demonstrate that once the HA fibers are removed, a scaffold that was 50% PCL, 25% PEO, and 25% HA would have infiltration attributes that are similar to a scaffold that has only 50% PCL and 50% PEO.

6. Fig 4c – how long was this assay (overnight or 24 hr)?

Response: These studies were carried out over 72 hours, with the scaffold incubated directly with the cartilage cylinders. To clarify this, we have also added this information to the figure caption.

7. Fig 6 – were the meniscus first soaked in tracker dye to distinguish it from mouse cells that could have migrated as well?

Response: Thank you for this comment. We did not label the meniscus implants in this study, but this is a good idea and will be carried in the future. That said, since the matrix formed was collagen II positive, it is very likely that most cells were derived from the implanted meniscus itself.

8. Fig 6 – Looks like five different statistical tests were performed on this data set. What were the statistical corrections that were made and how? In the Statistics section, it was noted that a Tukey’s HSD was performed. Was this done five times on the same data set?

Response: Thank you for this comment. We have clarified the statistical analysis. In this case both group and bin distance were used as factors in a 2 way ANOVA with Tukey’s correction.

9. Fig 7 – the meniscus contains both collagen I and II. Was collagen I examined?

Response: In response to this and the previous reviewer comments, we have added analysis of collagen I to the manuscript. Please see the new Figure 7c.

10. To give a clearer picture of biological variability and sample size, how many mice were used, and how many menisci (from different animals) were used? Samples per group were 4-5?

Response: This data is now included in the methods section. Please see page 16, lines 498 and 508.

11. Lines 219-221 – “This demonstrates that the scaffolds can exert a local effect on migration, while not compromising the overall integrity of the tissue.” With regard to overall integrity of the tissue, data showing repair or that the surrounding collagen has been damaged/are less dense and how deep this digestion occurs would support the hypothesis and allow a better evaluation of if the treatment is harmful. Supplementary Figure 4 shows something similar to what is requested, but the magnification is too low. Can zoomed-in images be provided of the interface and of the meniscus? What did the interface around the diameter of the scaffold (where it comes in contact with the rat) look like? Were there signs of enzymatic digestion?

Response: Thank you for this comment. Zoomed in images showing the interfaces and the internal structure of the meniscal explants are now provided (please see Supplementary Figures 6 and 7). Enzymatic digestion was minimal at the explant periphery.

12. Supplemental Figure 4 – in addition to zoomed-in images of the meniscus after collagenase digestion, please also provide zoomed-in images of the newly synthesized matrix in the empty and scaffolds.

Response: These images have been added. Please see Supplementary Figure 6a.

13. Fig 6d – what is the ROI for “scaffold center”? For example, is it 400-500 um from the interface of the scaffold and the tissue?

Response: Thank you for this comment. This data is now included in the figure legend and the methods have been revised to include this information as well.

14. For the pore size analysis, “areas <20 μm^2 were considered artifacts and excluded from analysis.” Were there more of these artifacts for one group/depth versus another? What did the data distribution look like?

Response: Thank you for this comment. We have reexamined this data and do not observe a difference in the number of small identified areas based on group or depth.

Reviewers' comments:

Reviewer #1 (Remarks to the Author):

The authors largely addressed my previous comments. Well done and thank you. This study really comes down to modulating pore size to allow for cellular infiltration. The authors have experience in this area, and even responded with data related to the average pore diameter and area fraction of fetal vs. adult tissue. However, this very relevant information was not included in the revised manuscript. By communicating a threshold pore size/area fraction for infiltration, the experimental findings become more significant because they can be integrated into experiments involving other dense-tissue systems beyond those studied here. These data should also inform the authors as to how much degradation is necessary in their scaffold (and the surrounding tissue) to allow good infiltration. Having high-quality, reliable, accurate porosity measures for each component in the study (transwells, fetal tissue, adult tissue, the various scaffolds) would provide what seems to be critical data. Distinguishing what measure – pore size, pore area, or some combination – is most important, and why, would also be helpful and is not currently included in the text.

Reviewer #2 (Remarks to the Author):

The Authors addressed all the comments constructively and the manuscript is now suitable for publication.

Response to Reviewers

Reviewer #1:

The authors largely addressed my previous comments. Well done and thank you.

Response: Thank you for your careful reading of our revised manuscript and positive comments. We have addressed the questions raised in the response below and in the revised manuscript. Thanks once again for your input towards improving this manuscript.

This study really comes down to modulating pore size to allow for cellular infiltration. The authors have experience in this area, and even responded with data related to the average pore diameter and area fraction of fetal vs. adult tissue. However, this very relevant information was not included in the revised manuscript. By communicating a threshold pore size/area fraction for infiltration, the experimental findings become more significant because they can be integrated into experiments involving other dense-tissue systems beyond those studied here.

Response: Thank you for this comment. We agree with the reviewer that it is beneficial to include information on tissue pore size, which has been added to the discussion section. Please see page 9, line 274.

These data should also inform the authors as to how much degradation is necessary in their scaffold (and the surrounding tissue) to allow good infiltration.

Response: Thank you for this comment. Our data suggests that a minimum of 10% area fraction of pores >5 μm diameter is required to encourage cell infiltration. This information has been added to the discussion section. Please see page 10, line 310.

Having high-quality, reliable, accurate porosity measures for each component in the study (transwells, fetal tissue, adult tissue, the various scaffolds) would provide what seems to be critical data.

Response: Thank you for this comment. We agree with the reviewer that it is useful to have porosity measures for each component in the study. However, since we used different methods to evaluate porosity (confocal microscopy for transwells and scaffolds, and second harmonic generation imaging for tissue), it is difficult to compare the absolute measures of pore size. In our case, we used these measures to assess relative changes to porosity compared to the appropriate controls in each experimental configuration.

Distinguishing what measure – pore size, pore area, or some combination – is most important, and why, would also be helpful and is not currently included in the text.

Response: Thank you for this comment. While both pore size and area fraction contribute to migration efficiency, our data and work by others show that pore size is ultimately the rate-limiting factor in migration. This information has been added to the discussion section. Please see page 9, line 280.

Reviewer #2:

The Authors addressed all the comments constructively and the manuscript is now suitable for publication.

***Response:** Thank you for your careful reading of our revised manuscript and recommending it for publication. We appreciate your input towards improving this manuscript.*